# Diode Laser Lithotription Technique Based on Optothermal Converter

**Olga S. Streltsova** [1], **Evgeny V. Grebenkin** [1,*], **Nikita M. Bityurin** [2], **Vladimir I. Bredikhin** [2], **Vadim V. Elagin** [3], **Vasily V. Vlasov** [1] and **Vladislav A. Kamensky** [3]

1   E.V. Shakhov Urology Department, Privolzhsky Research Medical University, 10/1 Minin and Pozharsky Square, 603005 Nizhny Novgorod, Russia; strelzova_uro@mail.ru (O.S.S.); urolog.75@mail.ru (V.V.V.)

2   Institute of Applied Physics, Russian Academy of Sciences, 46 Ulyanov St., 603155 Nizhny Novgorod, Russia; bit@appl.sci-nnov.ru (N.M.B.); bredikh@appl.sci-nnov.ru (V.I.B.)

3   Institute of Experimental Oncology and Biomedical Technologies, Privolzhsky Research Medical University, 10/1 Minin and Pozharsky Square, 603005 Nizhny Novgorod, Russia; elagin.vadim@gmail.com (V.V.E.); vlad@ufp.appl.sci-nnov.ru (V.A.K.)

\*   Correspondence: evgen-fifa@rambler.ru; Tel.: +7-(904)-3989-979

**Abstract:** Purpose: evaluation of the efficiency of the "hot spot" method for the fragmentation of urinary stones. Materials and methods: A retrospective analysis of clinical records of 1666 patients with urolithiasis who underwent percutaneous nephrolithoextraction/tripsy and contact ureterolithotripsy/ extraction in the period from 2014 to 2017 at the urology clinic was performed to assess the incidence of postoperative infectious and inflammatory complications. The research objects were postoperative urinary stones (*n*-78). The X-ray density and linear dimensions of the stones were determined. Stone fragmentation was performed with a continuous-wave diode laser operating at wavelengths of 0.81 μm, 0.97 μm, and 1.47 μm. An absorbing coating of micro-size graphite powder was applied on the working tip of the optical fiber. In vitro fragmentation was carried out in liquid. Results: A group of patients (224/1666) (13.4 ± 0.86%) was identified, who developed infectious and inflammatory complications after: percutaneous nephrolithotripsy, 123/361 (34.1 ± 2.5%) cases; percutaneous nephrolithoextraction, 59/240 (24.6 ± 2.78%); contact ureterolithotripsy, 23/294 (7.8 ± 1.57%); and ureterolithoextraction, 19/771 (2.5 ± 0.56%). In liquid, the "hot spot" technique made it possible to fragment stones with an X-ray density of up to 1000 HU at a laser wavelength of 0.81 μm, up to 1400 HU at 0.97 μm, and up to 1400 HU at 1.47 μm.

**Keywords:** micro size graphite optothermal convertor; hot-spot lithotripsy; diode lasers; urolithiasis; infection lithiasis

## 1. Introduction

Epidemiological data indicate a steady increase in the number of patients with urolithiasis worldwide [1]. Currently, the endoscopic laser lithotripsy is becoming the most widely used treatment for urolithiasis [2]. Holmium:YAG (Ho:YAG) laser lithotripsy allows to break up calculi of any chemical composition [3]. This method is based on water vaporization inside the stone, resulting in fragments with uncontrollable sizes. Recently, comparative data have been reported on three fundamentally different techniques for holmium laser lithotripsy, "dusting", "basketing", and "popcorning" [4–8]. However, these approaches are based on the explosive rupture of a stone. Dispersed fragments may lead to stone recurrence and inflammatory complications. In addition, during the operation, intrapelvic pressure increases, accompanied by pyelovenous and pyelotubular backflow. If the stones are infected, the spread of the fragments in the renal pelvicalyceal system can trigger postoperative inflammatory processes and then chronic inflammatory changes in the kidney. According to the scientific data, this is the cause of infectious and inflammatory complications in 30% of cases [9].

Thus, a technique is needed that would fragment potentially infected stones into pieces in a nonexplosive fashion and prevent them from spreading through the pelvicalyceal system. Our research group developed a "hot-spot" laser lithotripsy technique, which enables fragment calculi into two pieces using a fiber-coupled continuous-wave diode laser with a graphite microparticles-based optothermal converter on the working tip [10]. This converter has similar meaning to the blackening of the optical fiber tip [11–13]. Calculi are fragmented due to the high temperature of the distal tip of the optical fiber [14,15].

The "hot-spot" laser lithotripsy technique can reduce the scattering of stone fragments and affect pathogenic microorganisms in the contact zone due to the high temperature [16,17].

The aim of the study was to evaluate the frequency of infectious complications after traditional lithotripsy and to select the optimal laser wavelength for effective stone fragmentation through the "hot-spot" technique.

## 2. Materials and Methods

### 2.1. Retrospective Analysis

A retrospective analysis of the clinical records of 1666 patients with urolithiasis who underwent percutaneous nephrolithoextraction/nephrolithotripsy and contact ureterolithotripsy/ureterolithoextraction in the period from 2014 to 2017 was performed at the urology clinic of the Privolzhsky Research Medical University on the basis of the Nizhny Novgorod Regional Clinical Hospital named after N.A. Semashko. A total of 361 patients underwent percutaneous nephrolithotripsy (NLT), 240 patients underwent percutaneous nephrolithoextraction (NLE), 294 patients underwent contact ureterolithotripsy (ULT), and 771 patients underwent ureterolithoextraction (ULE). All patients were more than 18 years of age and provided written informed consent for the operation. The mean age of the patients was $54 \pm 2.3$ years. Surgical interventions were performed according to medical indications. A pneumatic lithotripter was mainly used for stone fragmentation. Percutaneous contact pneumatic lithotripsy was performed with the Karl Storz equipment, including a 28 Ch nephroscope and a 10 Ch rigid ureteroscope. A 30 Ch renal casing was used, which made it possible to minimize excess pressure in the pelvicalyceal system during the operation.

### 2.2. "Hot-Spot" Fragmentation Technique

To develop a stone fragmentation technique with minimal associated inflammation, we proposed to use a laser lithotripter based on a laser equipment for surgery and power therapy, LAKHTA-MILON (Milon Laser Co., Saint-Petersburg, Russia), with a light converter. The diode lasers are certified for medical use with fiber output on the same quartz fiber, were operated in a continuous mode of radiation generation. The laser radiation converter (LRC) is a layer of graphite powder (Figure 1c) that absorbs the visible and infrared light. The layer was formed at a fiber tip by applying of a colloidal solution of micro-size graphite powder in transparent silicone varnish, followed by drying with laser radiation (~0.1 W). When the layer is heated to 1000 °C, the silicone varnish decomposes, forming $SiO_2$, which serves as a binder in the coating. Figure 1 presents microphotographs of the tip face of the fiber with LRC (a, b) and of graphite powder particles (c). The size of graphite powder particles ranges from a few microns to tens of microns. Applying the LRC leads to absorbing 30–50% of the initial laser radiation. The graphite absorption coefficient in a wide wavelength range (including at $\lambda \approx 1$ μm) is known to be $10^6$ cm$^{-1}$ [18], therefore LRC transmission at 50–70% is determined by the heterogeneity of the graphite layer.

Thereby, 3–10 W of output laser power induces the LRC heating up to 2000 K [14]. The dependence of LRC temperature on ambient condition (air, vacuum and carbon dioxide) was presented in a recent work [10]. The carbon dioxide environment is preferred because oxygen contributes to the destruction of both the LRC and the quartz fiber. Permission was obtained from the Medical Ethics Committee of the Privolzhsky Research Medical University of the Ministry of Health of the Russian Federation (extract No. 8 dated 5

August 2019) to conduct a clinical study of the possibilities of applying the "hot-spot" continuous-wave diode laser technique to stone fragmentation in the urinary system.

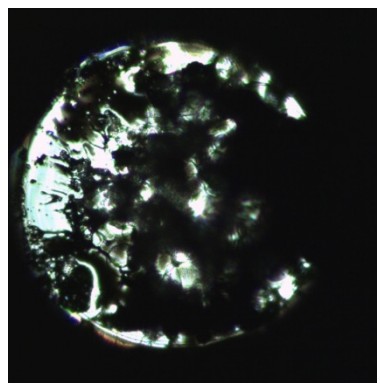 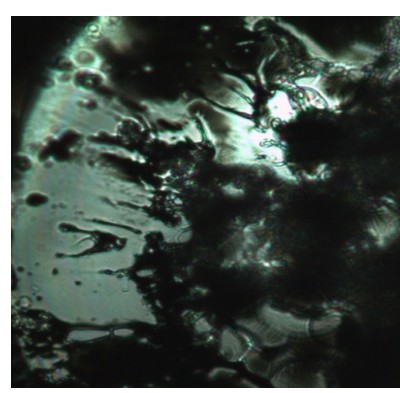 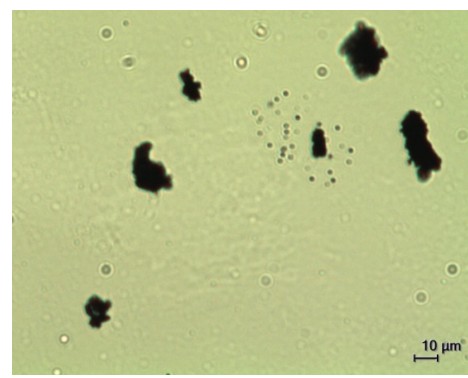

(**a**) Tip face of the fiber with LRC, 20×    (**b**) Tip face of the fiber with LRC, 60×        (**c**) Graphite, powder

**Figure 1.** Microphotographs: (**a**,**b**) view of the tip face of the fiber with LRC, (**c**) graphite powder particles.

### 2.3. Stones

The study was carried out in postoperative urinary stones (*n*-78). The stones were randomly divided into three groups, and their X-ray density was determined using a computed tomography in Hounsfield units (HU). A digital caliper was used to measure the maximum linear dimensions of a stone in the experiment. A stone chemical composition was determined using the traditional method of chemical analysis.

Stone fragmentation was performed with a continuous diode laser coupled with a 550-μm diameter fiber. Graphite carbon microparticles in silicone varnish were applied onto the working tip of the optical fiber. In the course of the study, the influence of the laser wavelength (0.81 μm, 0.97 μm, 1.47 μm), as well as the X-ray density of the stones on stone fragmentation in liquid, was investigated.

- Group 1—stone fragmentation at a wavelength of 0.81 μm in liquid (*n*-27);
- Group 2—stone fragmentation at a wavelength of 0.97 μm in liquid (*n*-36);
- Group 3—stone fragmentation at a wavelength of 1.47 μm in liquid (*n*-15).

### 2.4. Statistical Analysis

The obtained data were subjected to a statistical analysis using the Statistica 10 software package. The data of retrospective analysis of inflammatory complications were analyzed using Fisher's angular transformation method. The criterion assesses the significant difference between analyzing groups by the percent of inflammatory complications in each. The essence of Fisher's angular transformation consists in converting percentages to the values of the central angle, which is measured in radians. The results of measuring the linear dimensions of stones, their X-ray density, and fragmentation time are presented as the mean value and standard deviation. Differences were considered significant at $p < 0.05$.

## 3. Results

### 3.1. Evaluation of the Frequency of Infectious and Inflammatory Complications after Endoscopic/Percutaneous Operations for Urolithiasis

Based on the results of the laboratory blood and urine tests and an assessment of clinical examination data, a group of patients (224 people) (224/1666) (13.4 ± 0.86%) was identified, who developed infectious and inflammatory complications in the postoperative period characterized by fever and/or inflammatory changes in general blood and urine tests (Figure 2).

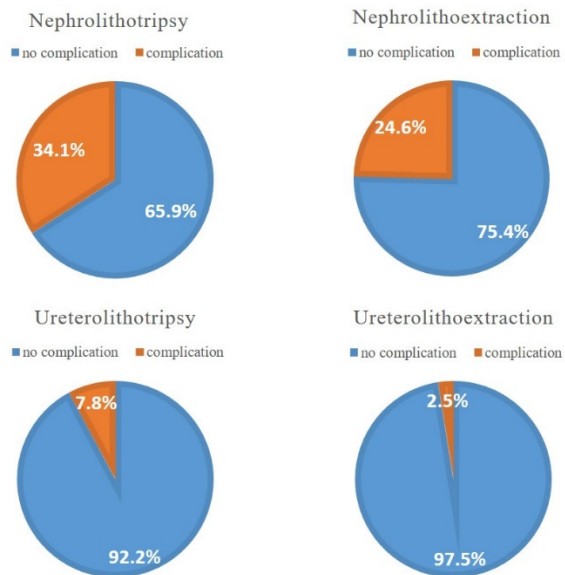

**Figure 2.** Percentage distribution of patients with infectious and inflammatory complications occurred in the postoperative period, and without them, depending on the type of endoscopic treatment of urolithiasis.

It was noted that in the percutaneous nephrolithotripsy group, postoperative infectious and inflammatory complications occurred in 123/361 (34.1 ± 2.5%) cases, while in the percutaneous nephrolithoextraction group this indicator was 59/240 (24.6 ± 2.78%). In the group of operations on the ureters, the proportion of patients with postoperative infectious and inflammatory complications after contact ureterolithotripsy was 23/294 (7.8 ± 1.57%), and after ureterolithoextraction was 19/771 (2.5 ± 0.56%).

There is a statistically significant difference at the $p < 0.001$ level, calculated by the Fisher angular transformation, between the incidence rates of complications for the various methods of endoscopic treatment used (Table 1).

**Table 1.** Significance levels ($p$) of differences in the incidence of complications in various types of endoscopic treatment.

| Operation | NLE | ULT | ULE |
|---|---|---|---|
| Nephrolithotripsy (NLT) | <0.01 | <0.001 | <0.001 |
| Nephrolithoextraction (NLE) | - | <0.001 | <0.001 |
| Ureterolithotripsy (ULT) | <0.001 | - | <0.001 |

In the nephrolithotripsy and nephrolithoextraction, the significance level of the difference is $p < 0.01$. Thus, it follows from the analysis that in terms of the occurrence of infectious and inflammatory complications in the postoperative period of endoscopic treatment of urolithiasis, the nephrolithotripsy technique is the least safe, and the safest technique is ureterolithoextraction.

### 3.2. Development of the Stone Fragmentation Methodology through the "Hot Spot" Method

The "hot-spot" fragmentation technique is relatively straightforward and involves the following steps. The stone was moved close to a bladder wall to prevent retropulsion. On the stone surface an imaginary line was drawn along which the stone should split. The fiber was placed on the line and then the laser emission was switched on. The stone fragmentation occurred in a drilling fashion and the laser fiber melted the stone (Figure 3). To prevent thermal injury, the laser emission was switched on during the drilling only. The fiber tip should remain in tight contact with the stone and sink into it during the fragmentation process. A few holes were drilled with a distance of 2 mm along the

imaginary line to fragment the stone. Finally, the stone was divided into two parts. Thus, the fragmentation time is the time required to divide the stone into two parts.

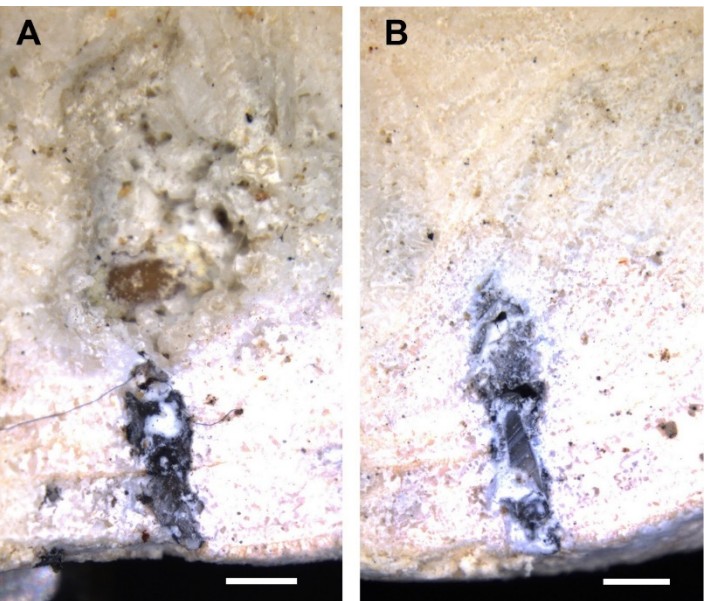

**Figure 3.** The longitudinal section of a hole melted in a stone by laser at 0.97 μm wavelength in air after 2 s (**A**) and 5 s (**B**). Scale bar is 1 mm.

The fragmentation of the stone was carried out in liquid. The average stone size was 10.26 ± 5.99 mm in the group with the laser wavelength of 0.81 μm, 12.5 ± 4.89 mm in the 0.97-μm laser group, and 8.4 ± 1.89 mm in the 1.47-μm laser group (Table 2). The average X-ray density was measured as 813.3 ± 479.88 HU, 1030 ± 426.8 HU and 932 ± 345.6 HU for the 0.81 μm, 0.97 μm and 1.47 μm groups, respectively. In the 0.81 μm laser group, all calculi with an X-ray density up to 1000 HU ($n = 18$) were successfully fragmented into two large fragments. Stones with a higher X-ray density in this group ($n = 9$), ranging from 1000 to 1735 HU, could not be fragmented. The maximum X-ray density of fragmented stones ($n = 27$) when fragmenting with a 0.97 μm laser was 1390 HU. The fragmentation of stones ($n = 9$) with a higher density (from 1401 to 1735 HU) was ineffective. The maximum X-ray density of stones ($n = 15$) fragmented with a 1.47 μm laser was 1390 HU. The analysis of the calculi chemical composition revealed that uric acid and struvite stones predominantly had an X-ray density of less than 1000. The most dense stones (over 1000 HU) were calcium oxalate. It was found that after making 15 holes, the drilling speed decreased and the converter must be reapplied.

**Table 2.** Stone parameters.

| Stone Parameters | 0.81 μm | 0.97 μm | 1.47 μm |
|---|---|---|---|
| $n$ | 27/9 * | 36/9 * | 15/0 * |
| Size, mm | 10.26 ± 5.99 | 12.5 ± 4.89 | 8.4 ± 1.89 |
| Mean fragmentation time, s | 16 ± 7.82 | 12.06 ± 4.42/31.27 ± 20.4 ** | 4.5 ± 1.32/19.14 ± 9.4 ** |
| Mean X-ray density, HU | 813.3 ± 479.88 | 1030 ± 426.8 | 932 ± 345.6 |
| X-ray density of non-fragmented stone, HU | >1000 ($n = 9$) | >1400 ($n = 9$) | — |

* The total number of stones/number of non-fragmented stones. ** The mean fragmentation time for stones with X-ray density up to 1000 HU/over 1000 HU.

### 3.3. Development of the Clinical Algorithm of the Urinary Stone Fragmentation

The following algorithm of the urinary stone fragmentation was developed using this technique.

(1) Insertion of sterile optical fiber (diameter 550 μm) into a ureteral catheter No. 6 Ch (the distal end is cut to move the optical fiber, in particular, into the bladder cavity);
(2) Install a rubber stopper at the proximal end of the optical fiber equal to the thickness of the stone (according to ultrasound and/or MSCT data), to prevent its distal end from being pulled out of the hollow guide tube more than the thickness of the stone;
(3) Install a catheterization cystoscope No. 25 Ch into the bladder, through the working channel of which the structure is introduced into its cavity;
(4) Press the optical fiber against the stone perpendicularly to the surface along the line of the planned fragmentation;
(5) Start the laser emission to create several channels-perforations along the fragmentation line;
(6) Extract the stone fragments with endoscopic forceps;
(7) Control examination of the bladder cavity and install a Foley catheter for one day.

According to the developed algorithm, ten operations of lithotripsy in the bladder cavity were successfully performed (Figure 4 A–C, Supplementary Video S1).

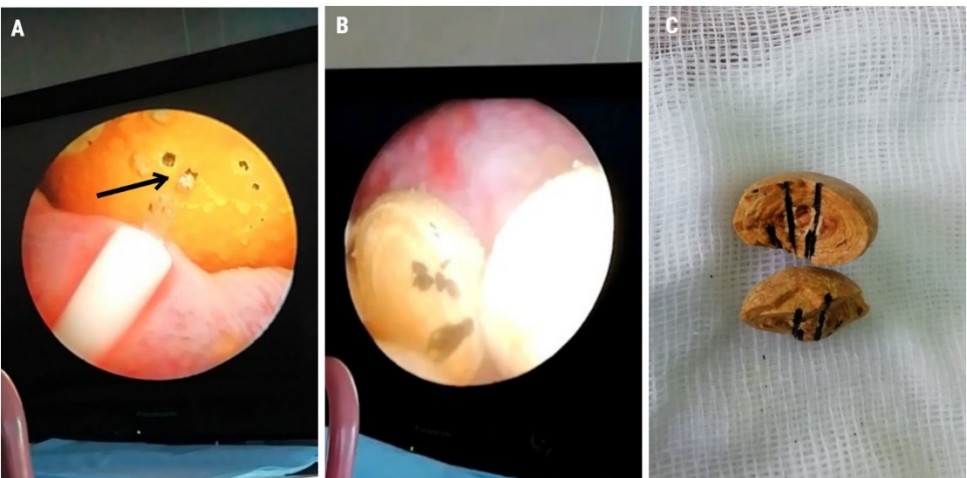

**Figure 4.** Controlled stone fragmentation in the operating room, stones in the bladder cavity (**A**–**C**): (**A**) the tip of the fiber (indicated by the arrow) is removed from the ureteral catheter (used as a casing) and brought to the stone; (**B**) one of the stones is fragmented into two fragments; (**C**) parts of the stone after extraction on the experimental table.

The laser-on time (tripsy time) was 15 to 33 s. At the same time, it takes up to 5 s to create each hole. The mean X-ray density of the stones was 300–420 HU for uric acid stones, 450–550 HU for magnesium ammonium phosphate stones, and 900–1040 HU for calcium phosphate stones. Visibility during the fragmentation was clear. No retropulsion was noted.

## 4. Discussion

The current paper presents a novel technique for urinary stone fragmentation to minimize the side effects of common lithotripsy approaches revealed during a retrospective study. The fine-sized fragmentation technique induced inflammatory complications. Taking into account the results of a retrospective analysis of complications after endoscopic operations, it was determined that the minimum number of complications is associated with the techniques of a whole stone extraction. To minimize the infection of the cavity system of the urinary organs, we proposed an improved "basketing" technique, where fragmentation is carried out by contact with the high-temperature tip of a silica fiber. This additionally has a bactericidal effect due to the high temperature, as shown earlier [16].

This method was used for porous stones containing biofilms, which were determined from the criterion calculi with an X-ray density up to 1400 HU [19].

The study of stone fragmentation in dependence of laser wavelength and their X-ray density is performed. The laser at 0.97 μm and 1.47 μm wavelengths demonstrated better effectiveness of stone fragmentation compared with the 0.81 μm laser wavelength. It is known that the coating applied onto a fiber tip absorbs only 30% of initial laser radiation [20]. The transmitted radiation may be absorbed by water in the stone, which leads to an increase in its temperature. The water absorption coefficient at 0.97 μm and 1.47 μm wavelengths is greater than that at the 0.81 μm wavelength (absorption coefficients are $0.02 \text{ cm}^{-1}$, $0.2 \text{ cm}^{-1}$ and $20 \text{ cm}^{-1}$ at a wavelength of 0.81 μm, 0.97 μm and 1.47 μm, respectively) [21].

It was shown that in liquid environment calculi with X-ray density up to 1000 HU fragmentation with the laser can be performed at 0.81 μm; while calculi with X-ray density up to 1400 HU, allows for fragmentation with the laser at 0.97 μm and 1.47 μm. In our previous study, experimental evaluations were carried out on model objects and on tissue of the ureter taken during autopsy, which showed the possibility of using this technique in clinical practice under certain exposure regimes. The following physical factors were considered as possible sources of injury during the operation: heating of the ureteral tissues adjacent to the stone to a traumatic value; and ignition of the organic coating of the optical fiber on the surface of the stone with the formation of a torch during the combustion reaction. The degree of trauma to the adjacent tissue was also assessed in case of a manipulation error during the operation, or short-term contact with the ureteral wall as a result of intraoperative discharge of the fiber from the stone surface [22]. Ho:YAG lithotripsy is known to break stones with various X-ray densities [23]. Our technique is applicable for potentially infected calculi. It is known that the infectious genesis of stones can be assumed from their X-ray density, since infected stones (struvite, apatite, ammonium urate stones, etc.) appear to have an X-ray density of less than 1400 HU [19]. Such types of stones typically have a porous structure, and the spread in their X-ray density can correspond to their mixed composition. According to the literature, infectious stones account for up to 51% of all urinary tract stones [24]. Eswara J.R. et al. (2013) retrospectively reviewed their experience of studying the flora associated with stones in patients undergoing ureteroscopic interventions. The authors found that urine cultures were positive in 7% of patients, while bacterial flora was isolated from calculi in 29% [25]. This shows the role played by the fine-size fragmentation of potentially infected kidney stones for the development of postoperative inflammatory response syndrome. The difficulty of treating patients with complicated urolithiasis and the inefficiency of antibacterial prophylaxis of postoperative inflammatory complications motivated this study.

Due to the physical characteristics of the stone fragmentation process, the "hot-spot" technique we applied enables stones to break into only two parts per one cycle. On one hand, it prevents a stone retropulsion as well as a dissemination of bacteria from calculi. In addition, due to the high temperature at the fiber tip, microorganisms on a fracture surface may be killed, producing the bactericidal effect of this technique [16]. On the other hand, this approach leads to an increase in the operation time, especially for large stones. Moreover, the optothermal converter should be renovated after 15 holes of drilling.

Based on obtained results of ex vivo stone fragmentation, the algorithm of the "hot-spot" technique for clinical application was developed. Ten operations using a 1.47 μm laser for this technique were conducted.

## 5. Conclusions

A retrospective analysis of clinical data showed that infectious and inflammatory complications are frequent in patients with urolithiasis after operations associated with crushing stones as well as in patients with infected lithiasis, which may be due to the release of bacteria, biofilms and their toxins associated with the stone.

The "hot-spot" technique based on diode lasers with a wavelength of 1.47 μm can be successfully used to fragment potentially infected stones with an X-ray density of up to 1400 HU. The algorithm of clinical application of the "hot-spot" technique allows for the reduction of complication frequency when tested on patients.

**Supplementary Materials:** The following are available online at https://www.mdpi.com/article/10.3390/photonics8100452/s1, Supplementary Video S1: Diode laser lithotripsy in the bladder.

**Author Contributions:** O.S.S.: Project development, Manuscript writing/editing. E.V.G.: Data collection, Data analysis, Manuscript writing. N.M.B.: Manuscript writing/editing. V.I.B.: Data analysis, Manuscript writing/editing. V.V.E.: Data collection, Data analysis, Manuscript writing. V.V.V.: Performing surgical operations, Data collection, Manuscript writing. V.A.K.: Project development, Manuscript writing/editing. All authors have read and agreed to the published version of the manuscript.

**Funding:** The work was financially supported by the Russian Science Foundation, grant No. 21-15-00371.

**Institutional Review Board Statement:** The study was conducted according to the guidelines of the Declaration of Helsinki, and approved by the Ethics Committee of FSBEI HE «Privolzhsky Research Medical University» MOH Russia (protocol code 8, date of approval 8 May 2019).

**Informed Consent Statement:** Written informed consent has been obtained from the patients to publish this paper.

**Data Availability Statement:** Not applicable.

**Conflicts of Interest:** The authors declare that they have no conflict of interest.

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
