# Peer review of "Diode Laser Lithotription Technique Based on Optothermal Converter"

_photonics, doi:10.3390/photonics8100452_

Round 1
Reviewer 1 Report
The authors presented a study that demonstrated a novel “hot-spot” lithotripsy technique using a diode laser with a light converter. The authors presented combined results of three different experiments:
- Based on a retrospective analysis of data authors evaluated that the inflammatory complications are associated with nephrolithotripsy;
- An in-vitro trial of the novel technique
- The authors presented an algorithm for urinary stones lithotripsy using the novel “hot-spot” technique and a preliminary clinical trial of using this technique in patients with bladder stones.
The topic of the trial is highly relevant. But, some issues should be addressed.
Major concerns:
- Materials and methods: As this is novel laser technology, so the main concern regarding its application on patients is safety. The authors indicated that the laser could heat up to 2000K which means 1726 °C. So, authors need to clarify how they evaluate whether the novel laser could lead to thermal damage of soft tissues of the pelvis, ureter and bladder or not.
- Results: authors proposed that the novel “hot-spot” technique could potentially reduce the scattering of the stones, however, did the authors evaluate a retropulsion rate during lithotripsy and did this technique allows reducing the retropulsion de facto comparing with conventional fragmentation technique using, for example, Ho:YAG?
- Results: As this technique requires lithotripsy in a contact mode “in a “drilling” fashion”, authors are encouraged to indicate a tip degradation rate of the fiber.
- Results: authors are encouraged to clarify what was defined as a “fragmentation time” and present data regarding laser-on time.
- As authors performed preliminary clinical trial, so they are encouraged to present data regarding this study (laser-on time, HU, stones composition, retropulsion and visibility during fragmentation)
- Authors are encouraged to indicate the limitations of their trials.
- The major concern in regards to the technique – in the current study authors evaluated large, soft stones and showed that extensive heating will ablate stone allowing them to rupture them. The finding seems to be far from a novelty. Available surgical lasers are also capable of achieving similar results, yet the reviewer believes that that would be inefficient. Therefore, the author encouraged them to report why they believe that technology could be effective in this regard?
Minor concerns:
- Introduction: authors are encouraged to shorten the introduction.
- Materials and methods: authors are encouraged to clarify the manufacturer of the laser that was used in the experiment.
- Extensive proofs reading is necessary.
Author Response
Dear Reviewer.
First of all, we would like to thank you for the very useful comments which allow improving the quality of the manuscript. We apologize very much for the delayed reply.
Please see the attachment.

Reviewer 2 Report
Introduction:
- The introduction is too long and needs to be revised.
- The authors should focus on laser lithotripsy used with f-URS and just explain how it works.
- - It is not necessary to mention SWL in the introduction.
Methods:
- What is the retrospective analysis ? This is not clear. If the stones used for the in vitro study are from these surgeries, this has to be mentioned. No need to mention the details of surgery, just "human stones were collected". Hounsfield units is not a good criterion. It would be better to use stone composition for the different groups.
- The design f the study is really not clear...
The manuscript must be revised ompletely because the design of the study is not clear. In my opinion, the authors should only focus on the "hot spot" technique. The retrospective analysis of charts to show the rate of infectious complications is not essential. Furthermore, instead of using stone density, it would have been better to use stone composition.
Author Response

(The authors gave the same response as above.)
